# PAX9 in Cancer Development

**DOI:** 10.3390/ijms23105589

**Published:** 2022-05-17

**Authors:** Xiaoxin Chen, Yahui Li, Chorlada Paiboonrungruang, Yong Li, Heiko Peters, Ralf Kist, Zhaohui Xiong

**Affiliations:** 1Cancer Research Program, Julius L. Chambers Biomedical Biotechnology Research Institute, North Carolina Central University, 700 George Street, Durham, NC 27707, USA; lchen@nccu.edu (X.C.); yli6@nccu.edu (Y.L.); chornarak@hotmail.com (C.P.); liyongdoctor@126.com (Y.L.); 2Department of Thoracic Surgery, National Cancer Center, Cancer Hospital of Chinese Academy of Medical Sciences, 17 Panjiayuan Nanli Road, Beijing 100021, China; 3Newcastle University Biosciences Institute, Newcastle upon Tyne NE2 4BW, UK; heiko.peters50@gmail.com; 4School of Dental Sciences, Newcastle University Centre for Cancer, Faculty of Medical Sciences, Newcastle University, Newcastle upon Tyne NE2 4BW, UK

**Keywords:** PAX9, cancer

## Abstract

Paired box 9 (PAX9) is a transcription factor of the PAX family functioning as both a transcriptional activator and repressor. Its functional roles in the embryonic development of various tissues and organs have been well studied. However, its roles and molecular mechanisms in cancer development are largely unknown. Here, we review the current understanding of PAX9 expression, upstream regulation of PAX9, and PAX9 downstream events in cancer development. Promoter hypermethylation, promoter SNP, microRNA, and inhibition of upstream pathways (e.g., NOTCH) result in PAX9 silencing or downregulation, whereas gene amplification and an epigenetic axis upregulate PAX9 expression. PAX9 may contribute to carcinogenesis through dysregulation of its transcriptional targets and related molecular pathways. In summary, extensive studies on PAX9 in its cellular and tissue contexts are warranted in various cancers, in particular, HNSCC, ESCC, lung cancer, and cervical SCC.

## 1. Introduction

Paired box 9 (PAX9) is a transcription factor of the PAX family. As a member of subgroup I, PAX9 protein contains an N-terminal DNA-binding paired box domain, an octapeptide, and a C-terminal transcription activation domain. The paired domain has two distinct helix-turn-helix motifs that interact with specific DNA sequences of target genes. The octapeptide may interact with other proteins. Sequence homology with potent transactivator proteins suggests a putative transactivation function at the C-terminus of PAX9. Yet, the functions of the octapeptide and the transactivation domain have not been well characterized [1,2].

Being expressed in pharyngeal pouches, limb, and craniofacial mesenchyme, PAX9 is essential for the development of the thymus, parathyroid, limb, palate, and teeth during mouse embryogenesis [3]. Subsequent studies have also demonstrated an essential role of *Pax9* in the development of filiform and taste papilla of the tongue, lip, intervertebral disc (in synergy with *Pax1*), and the cardiovasculature [4,5,6,7,8,9]. Homozygous *Pax9* knockout mice die shortly after birth [3], likely due to heart defects and inability to suckle and respiratory distress as a result of cleft palate and defects of the hyoid bone and thyroid cartilage [8]. In humans, patients with heterozygous mutations in *PAX9* commonly present with non-syndromic hypodontia or oligodontia [10], consistent with the gene dosage-dependent phenotype in mice [11].

In adult human tissues, PAX9 expression is restricted to the endocrine tissues (e.g., thymus, parathyroid), lymphatic system (tonsil), female reproductive system (vagina, cervix), and the upper aerodigestive tract (e.g., bronchus, tongue, esophagus, salivary gland, cheek epithelium) [12]. Low levels of expression are detected by single-cell RNAseq in the basal respiratory cells, club cells, ionocytes, respiratory ciliated cells, prostatic basal cells, prostatic glandular cells, and urothelial cells (Figure 1). Based on single-cell RNA expression data in human tissues (www.proteinatlas.org) (accessed on 22 March 2022), PAX9 is part of a single cell type expression cluster containing 176 genes (squamous epithelial cells-cornification) with high confidence (Appendix A), among which 15 are nearest neighbors: *ADH7, CALML3, CLCA2, DSG3, GBP6, KRT6C, KRT13, NMU, PITX1, PGLYRP3, RHCG, SERPINB13, SOX15, SPINK5,* and *UGT1A7.* Co-expression suggests a functional relationship between *PAX9* and these genes.

PAX genes are known to be involved in cancer. Interestingly, *PAX1/PAX9* and their related group *PAX4/PAX6* are believed to be neutral or tumor suppressors, whereas other *Pax* genes are believed to be oncogenic [1]. *PAX9* mRNA is differentially expressed in several human cancers as compared to the corresponding normal tissues (Figure 2). Genetic alterations (mutation, amplification, and deletion) of the *PAX9* gene are relatively common in lung adenocarcinoma and squamous cell carcinoma (SCC) (Figure 3) but rare in most other cancers. However, the mechanisms leading to these alterations and the carcinogenic mechanisms of these alterations remain unclear. Nevertheless, a high *PAX9* mRNA expression level is associated with a favorable prognosis of cervical and lung SCC (Figure 4).

The association between PAX genes and cancer and related molecular mechanisms has been reviewed in the literature [1,2,13,14,15,16,17]. This review summarizes the current understanding of PAX9 expression and functions in various cancers, pathways regulating PAX9 expression, and its downstream target genes and pathways. The available data suggest that overexpression or loss/mutation of PAX9 is unlikely to transform cells or induce cancer. In consideration of its role in the development of multiple tissues and organs, PAX9 likely regulates cell proliferation and differentiation of specific cells and thus contributes to cancer development in specific cell and tissue contexts when being silenced, mutated, or overexpressed. As a transcription factor, PAX9 can suppress or activate its downstream target genes and related molecular pathways, which may further promote or inhibit carcinogenesis in a context-dependent manner.

## 2. PAX9 in Specific Cancer

### 2.1. Head and Neck Squamous Cell Carcinoma (HNSCC)

In a study analyzing the data of gene expression microarrays and gene methylation microarrays, 15 differentially methylated genes were negatively associated with mRNA expression in HNSCC. Five aberrantly methylated genes (e.g., *PAX9*) were significantly correlated with overall survival [18]. However, in normal oral mucosa, eight genes, including *PAX9*, were consistently upregulated in oral mucosa and oral keratinocytes during wound healing [19].

### 2.2. Esophageal Cancer

PAX9 expression was either lost or significantly reduced in the majority of esophageal squamous cell carcinoma (ESCC) and squamous epithelial dysplasia (pre-cancerous lesion). The percentage of PAX9-positive cells within the esophageal epithelium decreased with increasing malignancy of the lesion [20]. Positive PAX9 expression has been correlated with a favorable postoperative prognosis and chemoradiosensitivity. In contrast, PAX9-negative ESCC had a significantly worse prognosis in disease-free survival and overall survival than PAX9-positive ESCC. The median overall survival for PAX9-negative ESCC was significantly shorter than that for PAX9-positive ESCC [21].

*Pax9* deficiency in the mouse esophagus promoted cell proliferation, delayed cell differentiation, and altered the global gene expression profile. When exposed to an esophageal carcinogen, *Pax9*-deficient mice developed significantly more tumors, dysplasia, and SCC in the forestomach than wild-type mice [22].

Interestingly, *PAX9* mutations were frequently seen in the histologically normal esophagus (HNE). Among five healthy organ donors aged between 85 and 93, four cases had *PAX9* mutations in HNE [23]. Moreover, *PAX9* mutations were significantly more frequent in HNE (*n* = 157) than in ESCC (*n* = 519), yet the distribution of *PAX9* mutation in HNE was similar to that in ESCC [24]. The frequency of *PAX9* mutation was not different between samples from high-risk individuals (individuals with a history of heavy smoking or drinking, *n* = 64) and samples from low-risk individuals (those with no history of heavy smoking or drinking, *n* = 93). Similar to *PAX9*, *NOTCH1* and *NOTCH* family genes were also much more frequently mutated in HNE than in ESCC. Using a diethylnitrosamine-induced mouse ESCC model, Colom et al. revealed a role for mutant cells in the elimination of less fit mutant cells and epithelial lesions. The survival of early tumors depended not only on their own mutations but also on the mutations in the neighboring HNE. These data suggest that some mutations found in HNE may contribute to the maintenance of homeostasis or even provide a protective effect [25]. On the contrary, other mutations (e.g., *TP53, NFE2L2, CDKN2A,* and *FBXW7*), which are more frequently observed in ESCC than in corresponding HNE, may be more positively selected than others during carcinogenesis. It is likely that *PAX9* mutant clones initially expand in histologically normal tissues, but most of these mutant clones are then eliminated during carcinogenesis and outcompeted by clones with functional *PAX9*. Positively selected clones in normal tissues may be further promoted by aging, chronic esophagitis, and environmental factors (e.g., alcohol drinking and tobacco). Such clones not only contribute to cancer development but also play a role in field cancerization through non-cell-autonomous effects [26].

As a well-established pre-cancerous lesion of esophageal adenocarcinoma, Barrett’s esophagus is characterized by intestinal metaplasia of the squamous epithelium. PAX9 was down-regulated in Barrett’s esophagus [27], suggesting its involvement in squamous differentiation and possible regulation of intestinal differentiation. In fact, during the development of the mouse esophagus, the expression of PAX9 and its downstream genes was associated with the terminal maturation of the squamous epithelium [28]. When a squamous transcription factor (P63) was knocked out and an intestinal transcription factor (CDX2) was overexpressed in the mouse esophagus, PAX9 was downregulated while metaplasia took place [29]. Similar to the mouse esophagus, morpholino knockdown of *pax9* in zebrafish resulted in loss or disorganization of the squamous epithelium of the upper digestive tract [30]. These data supported the hypothesis that PAX9 regulates squamous epithelial cell differentiation in the oro-esophageal epithelium.

### 2.3. Lung Cancer

High-resolution array analysis discovered a recurrent lung cancer amplicon located at 14q13.3. Fifteen percent of lung cancer samples had a low-level gain in this region, and another 4% had high-level amplification. Gene mapping revealed three genes (*TTF1/NKX2-1, NKX2-8, PAX9*) in the core region, all of which encode transcription factors involved in lung development. Amplification was also associated with gene overexpression. Overexpression of any pairwise combination of these genes in immortalized human lung epithelial cells had synergistic effects on promoting cell proliferation. Continuous expression of NKX2-8 and PAX9 was essential to the tumor maintenance of amplified SCC cells (H2170 cells). Experiments with both gene knockdown and overexpression further supported oncogenic roles for these genes [31]. These data suggest PAX9 may be a cell lineage dependency gene in certain lung cancers.

In lung SCC, *NKX2-1* loss rewired genomic occupancy of SOX2 and activated a squamous differentiation program. Interestingly, multiple SOX2-binding peaks specific to *NKX2-1^−/−^* cells appeared in the *SLC25A21* locus, and several regions in *SLC25A21* harbored functional *PAX9* enhancer activity in multiple species. Therefore, NKX2-1 loss and SOX2 overexpression drive the formation of lung SCC, likely through PAX9 in part [32].

Genome-wide CRISPR-Cas9 dropout screen in small cell lung cancer (SCLC) cells identified *PAX9* as an essential factor that is overexpressed and is transcriptionally driven by the BAP1/ASXL3/BRD4 epigenetic axis. PAX9 occupied distal enhancer elements and repressed gene expression by restricting enhancer activity. In multiple SCLC cell lines, *PAX9* deletion significantly induced a primed-active enhancer transition and caused overexpression of many neural differentiation and tumor-suppressive genes. Furthermore, PAX9 was found to interact and cooperate with the nucleosome remodeling and deacetylase complex at enhancers to repress nearby gene expression [33].

### 2.4. Cervical Cancer

PAX9 expression in cervical cancer tissue was lower than that in the adjacent normal tissues. It was correlated with the clinical stage, tumor size, infiltration depth, parametrium invasion, lymphovascular space invasion in tumor-positive lymph nodes, and prognosis. In cervical cancer cell lines (C-33A, CaSKi, HeLa, SiHa), the expression level of PAX9 was lower than that in normal cells (HCerEpiC). PAX9 overexpression inhibited the cancer cell proliferation and promoted apoptosis through the upregulation of caspase-3, poly (ADP-ribose) polymerase, and BAX and the down-regulation of BCL2. In vivo experiments demonstrated that PAX9 overexpression reduced the tumor weight and volume, decreased proliferation, and increased apoptosis [34]. An earlier study examined the anti-apoptotic roles of *PAX9* and *c-MYB* in KB cells, which were originally designated as oral cancer cells but are HeLa cells due to contamination. Inhibition of *PAX9* caused the induction of apoptosis with enhanced cleavage of caspase-3 and poly (ADP-ribose) polymerase, accelerated BAX, and reduced BCL2 expression. Moreover, *PAX9* siRNA arrested the cell cycle at the G_0_ phase [35]. These conflicting data, if both are true, may suggest that a low level of PAX9 expression in cervical cancer cells is essential for lineage survival, whereas a high level may suppress the cancer phenotype.

### 2.5. Ovarian Cancer

Soto et al. cross-analyzed data from methylome assessments and restoration of gene expression through microarray expression in a panel of four paired cisplatin-sensitive/cisplatin-resistant ovarian cancer cell lines. They also examined publicly available clinical data of the chemoresistant cases. *PAX9* was identified as a potential candidate gene, which exhibited epigenetic patterns of expression regulation. *PAX9* methylation was related to decreased overall survival in cisplatin-resistant patients. However, PAX9 overexpression did not affect cell survival [36]. Because *PAX9* mRNA and protein are expressed at an extremely low level in human ovarian tissue (Figure 1), it remains puzzling how and why PAX9 may play a critical role in ovarian cancer.

### 2.6. Breast Cancer

rs2236007 (*PAX9*) single nucleotide polymorphism (SNP) was found to be one of the 41 SNPs associated with breast cancer susceptibility [37]. Similarly, using a *cis*-expression quantitative trait loci analysis of normal and tumor transcriptome data, Guo et al. identified a list of 101 genes for 51 lead variants. Using luciferase reporter assays in ER^+^ MCF-7 cells (but not in ER^-^ SK-BR3 cells), alternative alleles of potentially functional SNPs, including rs2236007, significantly changed promoter activities of their target genes compared to reference alleles [38]. Multiple sequencing techniques further validated rs2236007 as a functional SNP in nine different breast cancer cell lines. The alteration at rs2236007 promoted the binding of a suppressive transcription factor EGR1 and resulted in *PAX9* downregulation. *PAX9* downregulation further promoted the cancer phenotype in vitro and was associated with a poor prognosis for breast cancer patients [39].

### 2.7. Chronic Lymphocytic Leukemia

In an analysis of chronic lymphocytic leukemia based on the mutational status of the immunoglobulin heavy chain variable gene (unmutated  =  39 vs. mutated  =  54), significantly higher *PAX9* mRNA expression was found in the unmutated subgroup. The relative risk of treatment initiation was significantly higher among patients with high expression of *PAX9* (RR  =  1.87, *p*  =  0.001). High expression of *PAX9* (HR: 3.14, *p*  <  0.001) was significantly associated with a shorter time to first treatment. The high expression of *PAX9* (HR: 3.29, *p*  =  0.016) was also predictive of shorter overall survival in patients with chronic lymphocytic leukemia [40]. Mechanistically, the role of PAX9 in leukemia may be due to its function in hematopoietic stem cell specification, likely through direct regulation of cytokine gene expression [41].

## 3. Upstream Regulation of PAX9 Expression

Regulation of PAX9 expression has been extensively studied in embryonic development, in particular, craniofacial and tooth development. Several molecular pathways and transcription factors, for example, GLI3 and SHH [42], BMP [43], SIX2 [44], and FGF [45], were found to control PAX9 expression during embryonic development. It remains unknown whether these molecular pathways or genes may regulate PAX9 expression in specific cancers. As mentioned above, the rs2236007 G allele in the *PAX9* promoter region increased EGR1 binding, which suppressed PAX9 expression in breast cancer cells [39]. The BAP1/ASXL3/BRD4 epigenetic axis promoted PAX9 expression in SCLC cells [33]. miR-130b, a highly expressed microRNA in ESCC, suppressed PAX9 expression in ESCC. The lncRNA DIO3OS upregulated PAX9 by binding to miR-130b, which ultimately promoted the radiosensitivity of ESCC in vitro and in vivo [46].

Promoter methylation has been shown as a regulatory mechanism of PAX9 expression in cancer. When we pyrosequenced human ESCC cells (KYSE70) treated with a demethylating agent (5-aza-2′-deoxycytidine), CpG sites in the promoter regions of two PAX9 transcriptional start sites became demethylated, and meanwhile, PAX9 expression was induced. In oral SCC tissue samples, the methylation levels of both PAX9 transcripts were higher in cancer than in matched normal tissues [22]. Similarly, in a recent study on tumor-specific DNA methylation in ESCC cases from nine high-incidence countries, the top three prioritized genes (*PAX9, SIM2,* and *THSD4*) shared similar methylation differences in the discovery and replication sample sets. These genes were exclusively expressed in normal esophageal tissues and downregulated in ESCC [47]. In addition to ESCC, promoter hypermethylation was also found in lung cancer cells, HNSCC, and ovarian cancer [18,36,48] (Figure 5) and associated with PAX9 expression and patient survival.

In our recent studies on alcohol-associated ESCC, ethanol exposure promoted carcinogen-induced oro-esophageal SCC in mice [22]. In order to understand the effects of ethanol on gene expression in esophageal squamous epithelial cells, gene microarray analysis showed downregulation of PAX9 target genes and RBPJ target genes in ethanol-exposed samples. Time- and dose-dependent exposure of human ESCC cells (KYSE510 and KYSE450) to ethanol confirmed down-regulation of PAX9 expression. When mice were exposed to ethanol in vivo, *Pax9* and its target genes were downregulated in the ethanol-exposed forestomach. Consistent with these data, PAX9 expression in the esophagus of drinkers was significantly lower than that of non-drinkers [49]. However, ethanol exposure of ESCC cells and the mouse forestomach did not induce *PAX9* promoter hypermethylation [22], suggesting alternative mechanisms responsible for PAX9 downregulation by ethanol exposure in our experimental settings.

We then further demonstrated the inhibition of the NOTCH pathway by ethanol exposure in vitro. The NOTCH pathway regulated PAX9 expression not only in human ESCC cells in vitro but also in the mouse esophagus in vivo. ChIP-PCR of two NOTCH factors (RBPJ and NICD1) confirmed *Pax9* as a direct downstream target of the NOTCH pathway. Furthermore, ethanol exposure inhibited the NOTCH pathway in the mouse esophagus and was associated with lower NICD1 expression in the human esophageal epithelium. These data supported a regulatory mechanism of PAX9 expression by the NOTCH pathway in the esophagus [49]. In fact, a genome-wide ChIPseq analysis of NICD1/RBPJ targets identified *PAX9* as a direct downstream target [50]. The NOTCH–PAX9 relationship is consistent with the fact that several members of the PAX family are also regulated by the NOTCH pathway, e.g., *PAX2* in the kidney [51], *PAX4* in blood [52], *PAX6* in the eye [53], and neuron [50], *PAX7* in muscle [54,55], and *PAX8* in otic placode [56] and thyroid [57].

## 4. Downstream Events of PAX9

As a transcription factor, PAX9 has an evolutionarily conserved paired domain that recognizes highly related DNA sequences. In the E12.5 wild-type mouse vertebral column, PAX9 ChIPseq revealed a binding motif (*5′-C/A G/A CGTGACCG-3′*) [2,7]. PAX9 also contains a conserved octapeptide motif, which functions as a transcriptional inhibitory motif. Thus, depending on the context and cofactors, PAX9 likely functions as either a transcriptional activator or a repressor [2,7].

In mouse embryonic fibroblasts, PAX9 as well as PAX3 function as redundant regulators of heterochromatin. They associated with DNA within pericentric heterochromatin and thus repressed RNA output from major satellite repeats. Simultaneous depletion of *Pax3* and *Pax9* resulted in dramatic derepression of major satellite transcripts, persistent impairment of heterochromatic marks, and defects in chromosome segregation. Methylated histone H3 at Lys9 was enriched at intergenic major satellite repeats only when the binding sites for PAX and other transcription factors remained intact. In addition, all histone methyltransferase Suv39h-dependent heterochromatic repeat regions in the mouse genome showed a high concordance with transcription factor binding sites [58]. PAX9 also behaves as an enhancer-specific binding factor [33]. In SCLC cells, <7% of PAX9 protein occupied the promoter/TSS regions at the chromatin. The vast majority of PAX9 occupied distal regulatory elements. Genetic deletion of *PAX9* led to increased expression of numerous neural differentiation genes and tumor suppressors genes, which is consistent with the SCLC phenotype.

During palatogenesis, PAX9 regulates multiple downstream factors, for example, *Msx1, Bmp4, Osr2, Fgf10,* and *Shh* [59,60]. It regulates the WNT pathway through transcriptional regulation of WNT pathway components, *Dkk1/Dkk2* and *Wnt9b/Wnt3*. Both small-molecule WNT agonists and genetic reduction of *Dkk1* corrected the cleft secondary palate in *Pax9*-deficient mice with the restoration of WNT pathway activities [61,62]. PAX9 also interacts with multiple pathways, for example, BMP/TGFβ pathways [7].

PAX9 regulates human ribosome biogenesis by acting as an RNA polymerase II transcription factor to influence the expression of multiple mRNAs required for pre-rRNA processing and global protein synthesis. Functionally, the phenotype in neural crest development due to *Pax9* deficiency was consistent with that found for the depletion of other ribosome biogenesis factors [63]. Interestingly, a broad role for dysregulated ribosome biogenesis has been suggested in the development and progression of most spontaneous cancers [64].

In *Pax9*-deficient esophagus, many genes associated with squamous cell differentiation (e.g., *Krtap3-3, Krt1-24/Krt35,* and *Sfrp5*) were significantly down-regulated. Meanwhile, SHH pathway genes (*Gli1* and *Gli2*), WNT pathway genes (*Wnt3* and *Gata3*), and stem cell markers (*Sox2* and *P63*) were upregulated. Gene set analysis showed enrichment of the SHH-signaling pathway, immune and inflammation pathways in *Pax9*-deficient esophagus, and enrichment of metabolism pathways and cell–cell junction pathways in wild-type esophagus [22]. In consideration of the important roles of the SHH pathway [65,66] and the WNT pathway [67,68] in ESCC, further studies are warranted to elucidate how *PAX9* downregulation may contribute to ESCC through these molecular pathways.

## 5. Future Directions

As of March 2022, only 449 “PAX9” manuscripts and 63 “PAX9 and cancer” manuscripts have been published since 1993 in the Pubmed database (https://pubmed.ncbi.nlm.nih.gov/) (accessed on 22 March 2022). Most of these manuscripts are related to the functions of PAX9 and related diseases in craniofacial development. Therefore, the role of PAX9 in cancer development is largely unknown. Although PAX9 is often called an oncogene or tumor suppressor gene in the literature, neither overexpression nor loss/mutation of PAX9 is sufficient to induce cancer.

Cellular and tissue contexts seem to be important for the functions of PAX9 in cancer development. PAX9 downregulation could make normal cells susceptible to neoplasia, whereas, when cancer has developed, *PAX9* amplification and overexpression could also promote the cancer phenotype (e.g., HNSCC, ESCC, and lung cancer), similar to the case of NOTCH in carcinogenesis [69].

The differential expression pattern of PAX9 in cancer and normal tissues may suggest its involvement in carcinogenesis. PAX9 is upregulated in three bone or bone marrow cancer types, T-cell acute lymphocytic leukemia, Ewing sarcoma, and osteosarcoma [2]. It would be interesting to examine whether PAX9 upregulation may contribute to these cancers. As an essential gene for the development of the thymus and parathyroid, PAX9 may also play a role in thymoma and thymic carcinoma and parathyroid cancer.

In conclusion, understanding the roles and molecular mechanisms of PAX9 in cancer development will be a fruitful research area. Studies on PAX9 are needed in various cancers, in particular, HNSCC, ESCC, lung cancer, and cervical SCC (Figure 6). Several mouse lines are available for studies on cancer development in vivo: *Pax9* hypomorphic mouse [11], *Pax9* knockout mouse [3], *Pax9^fl/fl^* mouse [70], *Pax9CreER* mouse [71], and *Pax9Cre* mouse [9]. In addition to mice, zebrafish are also suitable models for cancer research. In the literature, zebrafish have been used successfully to elucidate the role of *pax9* in the development of blood cells [41,72], oro-esophageal squamous epithelium [30], fin [73], tooth [74], and palate [75].

## Figures and Tables

**Figure 1 ijms-23-05589-f001:**
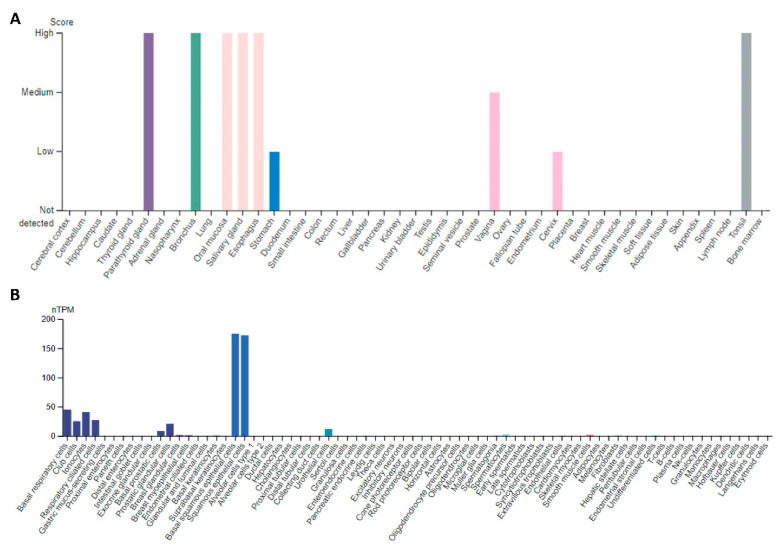
PAX9 expression in human tissues and cells (www.proteinatlas.org) (accessed on 22 March 2022). (**A**) PAX9 protein expression based on immunostaining of tissue sections; (**B**) *PAX9* mRNA expression based on single-cell RNAseq data.

**Figure 2 ijms-23-05589-f002:**
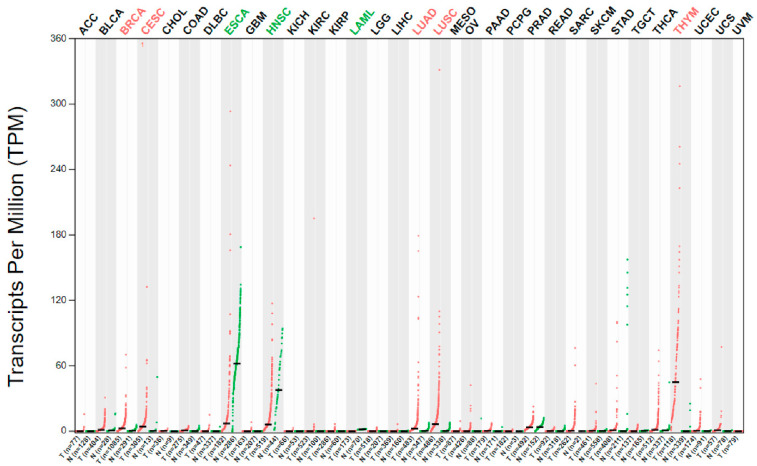
Differential expression of *PAX9* mRNA in cancer and corresponding normal tissues (http://gepia.cancer-pku.cn/) (accessed on 22 March 2022). RED indicates significant *PAX9* overexpression in BRCA, CECS, LUAD, LUSC, and THYM, and GREEN significant downregulation in ESCA, HNSC, and LAML, as compared to corresponding normal tissues. T—tumor tissue; N—normal tissue. Abbreviations: ACC—adrenocortical carcinoma; BLCA—bladder urothelial carcinoma; BRCA—breast invasive carcinoma; CESC—cervical squamous cell carcinoma and endocervical adenocarcinoma; CHOL—cholangiocarcinoma; COAD—colon adenocarcinoma; DLBC—lymphoid neoplasm diffuse large B-cell lymphoma; ESCA—esophageal carcinoma; GBM—glioblastoma multiforme; HNSC—head and neck squamous cell carcinoma; KICH—kidney chromophobe; KIRC—kidney renal clear cell carcinoma; KIRP—kidney renal papillary cell carcinoma; LAML—acute myeloid leukemia; LGG—brain lower grade glioma; LIHC—liver hepatocellular carcinoma; LUAD—lung adenocarcinoma; LUSC—lung squamous cell carcinoma; MESO—mesothelioma; OV—ovarian serous cystadenocarcinoma; PAAD—pancreatic adenocarcinoma; PCPG—pheochromocytoma and paraganglioma; PRAD—prostate adenocarcinoma; READ—rectum adenocarcinoma; SARC—sarcoma; SKCM—skin cutaneous melanoma; STAD—stomach adenocarcinoma; TGCT—testicular germ cell tumors; THCA—thyroid carcinoma; THYM—thymoma; UCEC—uterine corpus endometrial carcinoma; UCS—uterine carcinosarcoma; UVM—uveal melanoma.

**Figure 3 ijms-23-05589-f003:**
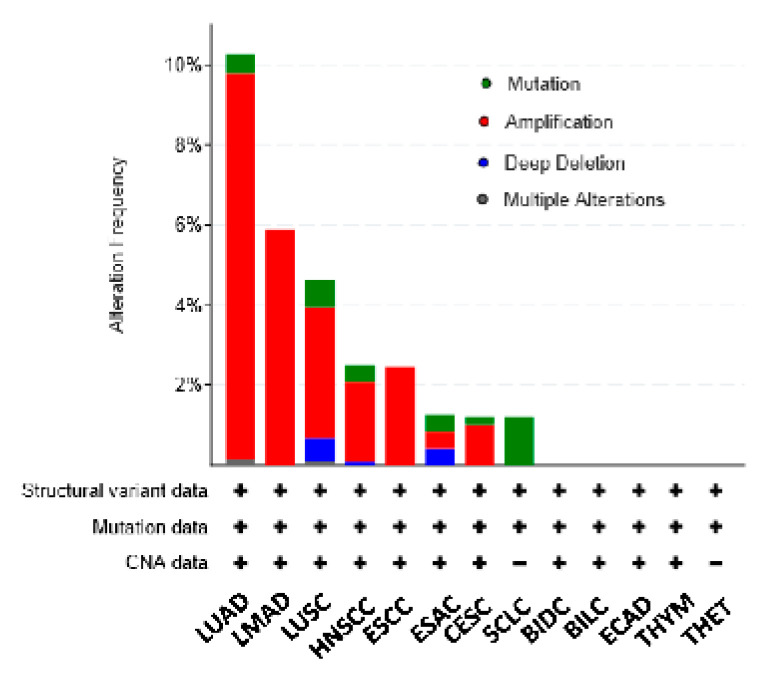
Genetic alterations of the *PAX9* DNA in human cancers that differentially express PAX9 (www.cbioportal.org) (accessed on 22 March 2022). *PAX9* amplification may lead to overexpression that promotes lung cancer development. LUAD—lung adenocarcinoma; LMAD—lung mucinous adenocarcinoma; LUSC—lung squamous cell carcinoma; HNSCC—head and neck squamous cell carcinoma; ESCC—esophageal squamous cell carcinoma; ESAC—esophageal adenocarcinoma; CESC—cervical squamous cell carcinoma; SCLC—small-cell lung carcinoma; BIDC—breast invasive ductal carcinoma; BILC—breast invasive lobular carcinoma; ECAD—endocervical adenocarcinoma; THYM—thymoma; THET—thymic epithelial tumor.

**Figure 4 ijms-23-05589-f004:**
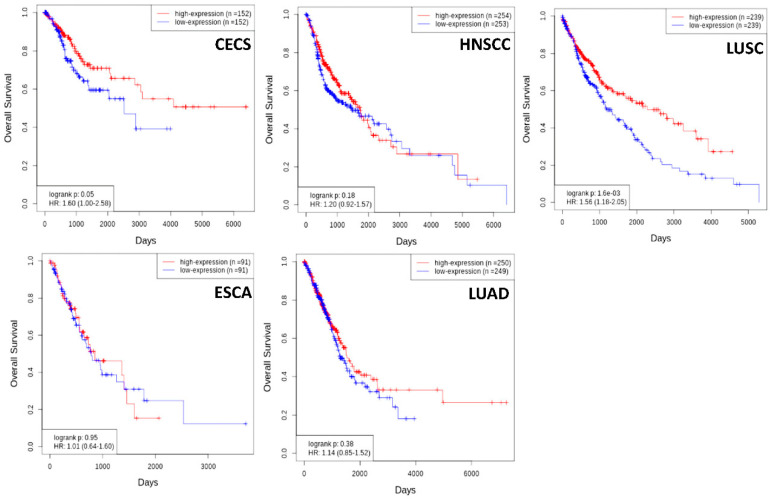
High expression of *PAX9* mRNA is associated with a favorable prognosis in cervical squamous cell carcinoma (CECS) and lung squamous cell carcinoma (LUSC), but not in head and neck squamous cell carcinoma (HNSCC), esophageal carcinoma (ESCA), and lung adenocarcinoma (LUAD) (www.oncodb.org) (accessed on 22 March 2022).

**Figure 5 ijms-23-05589-f005:**
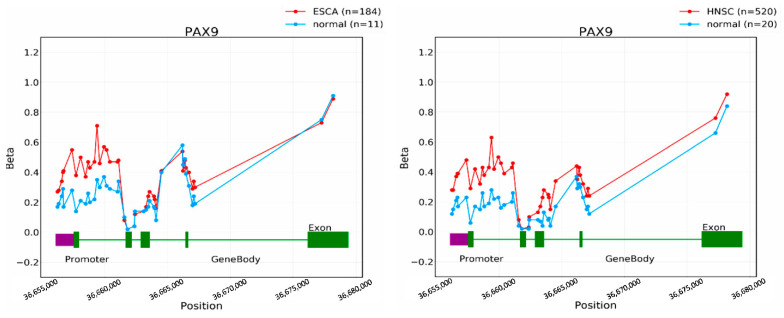
*PAX9* promoter hypermethylation in esophageal carcinoma (ESCA) and head and neck squamous cell carcinoma (HNSCC) (www.oncodb.org) (accessed on 22 March 2022). The beta value indicates the methylation level.

**Figure 6 ijms-23-05589-f006:**
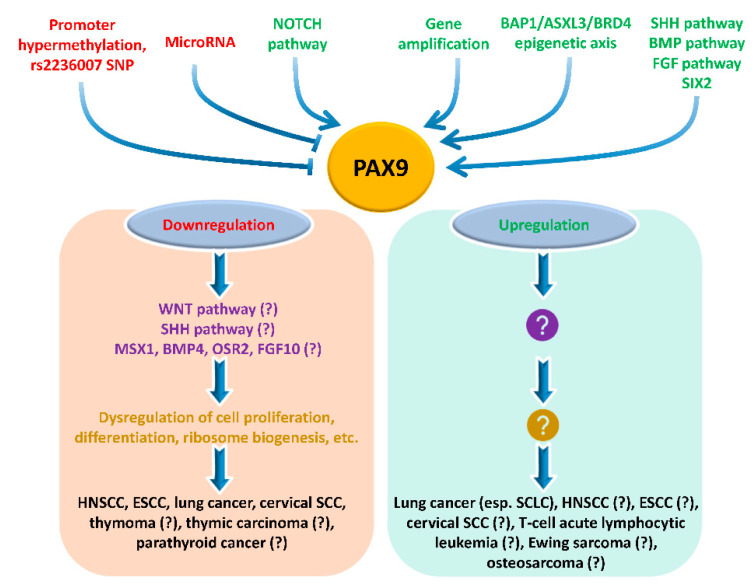
The role of PAX9 in cancer development. PAX9 expression is regulated by multiple upstream regulators. Aberrant expression of PAX9 impacts downstream genes and pathways and cellular phenotypes. However, its functions in cancer development depend on the cellular and tissue context, and its mechanistic roles in specific pathways, cellular phenotypes, and human cancer are largely unknown (question marks). ←, activation; ˫, inhibition.

## Data Availability

Not applicable.

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
