# Peer review of "PAX9 in Cancer Development"

_ijms, 2022, doi:10.3390/ijms23105589_

Round 1

Reviewer 1 Report

Reviewer’s Report:

The current review, “PAX9 in Cancer Development” by Chen X and colleagues sincerely demonstrated the molecular role of PAX9 proteins in cancer, starting from a general expression pattern of PAX9 in different human tissues, as well as, in the different cancer types, molecular functions of PAX9 known till date and finally, certain upstream, and downstream regulators of PAX9 in an efficient way that might be beneficial to the general readers. A couple of major points that need to be addressed for its betterment are mentioned below,

  1. The structural details of PAX9 protein, including domain specificity along with the functional relevance of its various domains, need to be incorporated just after the introduction section.
  2. All known upstream and downstream regulators of PAX9 need to be enrolled in a tabular form including major pathway/signaling cascade related to, cancer type associated with, and the relevant reference info must have to be incorporated in a table for better visibility towards the general readers.

Reviewer 2 Report

I read with interest the manuscript Review entitled " PAX9 in Cancer Development " that has been submitted to IJMS Journal.

This Review collects and properly summarizes the most significant data on the subject. 

- The review significantly contributes to the field of PAX9 function in cancer

- The work is well organized and comprehensively described

- The review is scientifically sound

- The cited references are appropriate and updated

In my opinion, this paper is acceptable in IJMS journal.

Reviewer 3 Report

This is an interesting review regarding the role of Pax9 in cancer. I have no major comment but I would suggest splitting fig 4 a, b, c d and e

Reviewer 4 Report

This is quite interesting review about the role of PAX9 in cancer. Most important and significant, on my oppinion, are parts 3, 4 and 5. At the same time, parts 1 and 2 are quite poorly written in a style "all-what-I-know", i.e., they are lacking good structure and appropriate analysis. Figures 2, 3 and 4 seems somehow confusing without additional comments For example, fig. 2 shows downregulation of the gene expression in ESCA, but figure 3 shows amplpification of the gene in ESCC and ESAC. Again, figure 2 shows overexpression of the gene in CESC and better survival is shown in fig. 4 in relation with higher expression rates of the gene in patients with this type of cancer. In part 2, presentation of the different findings is quite chaotic - it is obvious, that different changes in PAX9  (SNPs, mutations, ampification, hyper- or hypomethylation) may have very different consequences, but all they are discussed in a single paragraph.

Overall, my recommendation is to exclude figures 2, 3 and 4 together with a corresponding text and to correct part 2. Probably, this part could be replaced by the table summarizing different findings in different cancers accompanied by short comments. 

Additional remarks:

  1. Please, identify in fig. 5 what does "beta" mean?
  2. Lines 296-297, unpublished data are cited. Are these data collected by the authors of the manuscript? Is it possible to give more detailed information about these findings?
